# Lower Geriatric Nutritional Risk Index and Prognostic Nutritional Index Predict Postoperative Prognosis in Patients with Hepatocellular Carcinoma

**DOI:** 10.3390/nu16070940

**Published:** 2024-03-25

**Authors:** Mariko Tsukagoshi, Kenichiro Araki, Takamichi Igarashi, Norihiro Ishii, Shunsuke Kawai, Kei Hagiwara, Kouki Hoshino, Takaomi Seki, Takayuki Okuyama, Ryosuke Fukushima, Norifumi Harimoto, Ken Shirabe

**Affiliations:** Division of Hepatobiliary and Pancreatic Surgery, Department of General Surgical Science, Gunma University Graduate School of Medicine, 3-39-22 Showa-machi, Maebashi 371-8511, Gunma, Japan; marikot@gunma-u.ac.jp (M.T.); kshirabe@gunma-u.ac.jp (K.S.)

**Keywords:** geriatric nutritional risk index, hepatocellular carcinoma, liver resection, prognosis, prognostic nutritional index, recurrence

## Abstract

Increasing evidence suggests that nutritional indices, including the geriatric nutritional risk index (GNRI) and prognostic nutritional index (PNI), are predictors of poor prognosis in patients with hepatocellular carcinoma (HCC). Hence, this study aimed to explore the value of the GNRI and PNI in evaluating postoperative prognosis in patients with HCC, particularly regarding its recurrence patterns. We performed a retrospective analysis of 203 patients with HCC who underwent initial hepatic resection. Patients were divided into two groups according to the GNRI (cutoff: 98) and PNI (cutoff: 45). The GNRI and PNI were significantly associated with body composition (body mass index and skeletal muscle mass index), hepatic function (Child-Pugh Score), tumor factors (tumor size and microvascular invasion), and perioperative factors (blood loss and postoperative hospitalization). Patients with a low PNI or low GNRI had significantly worse overall survival (OS) and recurrence-free survival. Patients with early recurrence had lower PNI and GNRI scores than those without early recurrence. Patients with extrahepatic recurrence had lower PNI and GNRI scores than those without extrahepatic recurrence. The PNI and GNRI might be useful in predicting the prognosis and recurrence patterns of patients with HCC after hepatic resection.

## 1. Introduction

Hepatocellular carcinoma (HCC) is the sixth most common cancer and the third leading cause of cancer-related death worldwide [1]. Hepatic resection is one of the mainstay curative treatments for HCC. Recent advances in perioperative management and strict surgical criteria have enabled safe hepatic resection in patients with HCC. However, patients with HCC show high recurrence rates even after curative surgical resection, and many cases develop an unresectable status [2]. For precise treatment and improvement in patient prognosis, early identification of patients at a high risk of recurrence and timely individualized therapeutic strategies are crucial.

Malnutrition is a common and serious problem in patients with HCC. Several studies have shown that preoperative nutritional status is associated with poor prognosis in patients with HCC [3,4,5]. Recently, several biomarkers of nutritional status, notably the prognostic nutritional index (PNI) and geriatric nutritional risk index (GNRI), have been studied for their prognostic role in patients with HCC. The PNI is a marker that is calculated based on serum albumin levels and lymphocyte counts. The PNI was initially used to assess the immunological and nutritional conditions of patients with digestive diseases and is shown to be a prognostic marker for patients with gastrointestinal malignancies, including HCC [6,7]. The GNRI has been proposed to evaluate nutrition-related risks in elderly patients and can be easily calculated based on body weight, height, and serum albumin levels [8]. Many studies have found that the GNRI can be used to assess the prognosis of various malignant tumors, including HCC [9,10,11,12]. Recently, Yang et al. reported that a combination of the GNRI and PNI could distinguish between the risks of overall survival (OS) and recurrence-free survival (RFS) after surgery in patients with HCC [13].

Although the PNI and GNRI have been shown to be associated with prognosis in patients with HCC after liver resection, there are still few reports on their relationship with recurrence patterns. Therefore, we conducted a retrospective study of patients with HCC undergoing surgical resection to investigate the prognostic impact of the PNI and GNRI and to reveal their relationship with postoperative recurrence patterns.

## 2. Materials and Methods

### 2.1. Patient Selection

We retrospectively analyzed 203 consecutive patients with HCC who underwent initial hepatic resection between January 2016 and September 2022 at the Department of Hepatobiliary and Pancreatic Surgery, Gunma University Hospital. This study was approved by the ethics committee of the hospital (HS2021-190). All clinical samples were used in accordance with the institutional guidelines and the Declaration of Helsinki after obtaining signed informed consent from all participants.

### 2.2. Data Collection and Treatment

The baseline clinical and demographic characteristics and treatment-related details of all patients were collected from their medical records. Positive anti-HCV findings were considered to show that HCC was caused by hepatitis C virus (HCV), whereas HCC due to hepatitis B virus (HBV) was determined when the HBV surface antigen was positive. Surgical procedures were performed according to the institutional policies and institutional cancer board recommendations. The skeletal muscle (SM) mass area at the inferior aspect of the third lumbar vertebra (L3) was measured using computed tomography (CT). Muscle area was normalized as follows: SM index (SMI) = cross-sectional area of the SM in the L3 region/height^2^ (cm^2^/m^2^). Postoperative complications within 30 days were scored according to the Clavien–Dindo classification [4]. The resected tumors were classified according to the TNM Classification of Malignant Tumors of the Union for International Cancer Control (8th version). RFS was defined as the period from the date of surgery to that of documented recurrence or all-cause death. OS was defined as the period from the date of surgery to the date of all-cause death.

### 2.3. Definition of the GNRI and PNI

Preoperative nutritional status was assessed using the GNRI and PNI. According to the literature, the GNRI and PNI are calculated as follows: GNRI = [14.89 × serum albumin [g/L]) + (41.7 × actual/ideal body weight [kg]). Ideal body weight was calculated as follows: ideal body weight = patient’s height (m) × height (m) × 22 (body mass index [BMI]). When the actual preoperative body weight was higher than the ideal weight, the ratio was set to 1. Based on previous research, a GNRI < 98 was considered low, and a GNRI ≥ 98 was considered a high level. The PNI formula is as follows: PNI = 10 × serum albumin (g/dL) + 0.005 × total lymphocyte count (/mm^3^); a PNI < 45 was considered low, and PNI ≥ 45 was considered a high level [14]. Using these indices, a PNI ≥ 45 and a GNRI ≥ 98 were defined as a normal nutritional status.

### 2.4. Follow-Up

All patients were examined every 3 months for recurrence after discharge using tumor markers and CT or magnetic resonance imaging. Recurrent HCC is treated with surgery, chemotherapy, transcatheter arterial chemoembolization, radiotherapy, or heavy ion radiotherapy, depending on the recurrence situation.

### 2.5. Statistical Analysis

Categorical variables were assessed using the chi-square test or Fisher’s exact test, as appropriate. The Mann–Whitney *U* test was used to analyze continuous variables. Survival curves were estimated using the Kaplan–Meier method, and the log-rank test was used to analyze the differences between the curves. Cox proportional hazards model analysis was performed using univariate and multivariate analyses of prognostic factors. All statistical analyses were performed using the JMP Pro 14 software (SAS Institute, Cary, NC, USA). Values of *p* < 0.05 were considered to indicate statistical significance.

## 3. Results

### 3.1. Clinical Characteristics of Patients in the Two Groups Classified According to the PNI and GNRI

A comparison of the demographic and clinical characteristics between the two groups classified according to the PNI or GNRI is shown in Table 1. Based on this definition, 45 (22%) patients were assigned to the low PNI group and 34 (17%) patients were assigned to the low GNRI group. Basal liver disease was due to HCV in 78 patients (38.4), HBV in 22 (10.8%), and other causes in 103 (50.7%) patients. Of the 78 patients with HCV, 54 (69.2%) patients had obtained a sustained virological response due to antiviral treatment (interferon or direct acting antiviral treatment) before the initial hepatic resection.

The low PNI group was significantly correlated with a lower BMI, SMI, and hand grip strength. Significant associations were observed between a low PNI and preoperative blood data indicating a lower lymphocyte count, lower albumin levels, higher C-reactive protein (CRP) levels, a lower Child-Pugh Score A, and higher AFP levels. Additionally, a low PNI was significantly associated with a longer operation time, more blood loss, longer postoperative hospitalization, a larger tumor size, and greater microvascular invasion. In contrast, the GNRI was significantly correlated with a lower BMI, lower SMI, lower albumin levels, higher CRP levels, a lower Child-Pugh Score A, more blood loss, longer postoperative hospitalization, and a larger tumor size. No statistically significant associations were observed between a low PNI or GNRI and postoperative complications.

### 3.2. Association between Nutritional Index and Prognosis

The prognostic significance of the PNI and GNRI is shown in Figure 1. Patients with a low PNI had significantly worse OS (Figure 1a) and RFS (Figure 1b) than those with a high PNI. Similarly, patients with low GNRI scores had significantly worse OS (Figure 1c) and RFS (Figure 1d) than those with high GNRI scores. Based on both definitions, 55 (27%) patients had a low PNI or a low GNRI. Patients with a low PNI or a low GNRI had significantly worse OS (Figure 1e) and RFS (Figure 1f) than those with a normal nutritional status (PNI ≥ 45 and GNRI ≥ 98).

### 3.3. Prognostic Factors Associated with OS

Univariate and multivariate analyses were performed to analyze factors associated with OS (Table 2). The univariate analysis showed that male sex, an ICG-R15 > 10%, a PNI < 45, a GNRI < 98, an operation time > 300 min, blood loss > 500 mL, complications, a tumor size > 30 mm, and microvascular invasion were significantly associated with reduced OS. The multivariate analysis demonstrated that male sex, an ICG-R15 > 10%, a tumor size > 30 mm, microvascular invasion, and a PNI < 45 or a GNRI < 98 (hazard ratio [HR] = 2.21; 95% confidence interval [CI]: 1.21–4.05; *p* = 0.010) were independent prognostic factors for OS.

A subgroup analysis was performed to clarify the impact of the PNI and GNRI in patients with liver dysfunction (ICG-R15) and patients with advanced tumors (a larger tumor size and with microvascular invasion). A low PNI or GNRI was an independent risk factor for OS in the subgroup analysis of the patients with an ICG R15 > 10%, with a tumor size > 30 mm, and without microvascular invasion (Table 3). In the subgroup analysis of the patients with an ICG R15 ≤ 10%, there was no significant factor for OS in the univariate analysis. Furthermore, in the subgroup analysis of the patients with a tumor size ≤ 30 mm, microvascular invasion was the only independent prognostic factor in the univariate analysis for OS.

### 3.4. Prognostic Factors Associated with RFS

The univariate and multivariate analyses of RFS factors are shown in Table 4. The univariate analysis revealed that male sex, a PNI < 45, a GNRI < 98, blood loss > 500 mL, a tumor size > 30 mm, and microvascular invasion were significantly associated with reduced RFS. The multivariate analysis demonstrated that male sex and microvascular invasion were independent prognostic factors of RFS.

### 3.5. Correlation between Nutritional Index and Recurrence Timing and Pattern

During a median follow-up of 39.2 months, a total of 40 (72.7%, low-PNI or low-GNRI group) and 74 (50.0%, normal nutritional status group) patients experienced HCC recurrence. We investigated the association between nutritional index and recurrence timing and pattern. Early (within one year after surgery) recurrence was significantly higher in patients with a low PNI or GNRI (49.1%) compared to patients with a normal nutritional status (28.4%) (*p* = 0.008) (Figure 2a). Both the PNI and GNRI were significantly lower in patients with early recurrence (Figure 2b,c). Extrahepatic recurrence was significantly more common in patients with a low PNI or GNRI (23.6%) than in patients with a normal nutritional status (8.8%) (*p* = 0.008) (Figure 3a). Both the PNI and GNRI were lower in patients with extrahepatic recurrence (Figure 3b,c).

## 4. Discussion

The present study focused on the impact of the PNI and GNRI in patients with HCC who underwent surgical resection. This report showed that the PNI and GNRI were significantly associated with body composition, hepatic function, tumor factors, and perioperative factors. In addition, a low PNI or GNRI was an independent prognostic factor for OS in patients with HCC undergoing surgical resection. Moreover, patients with a low PNI or GNRI had significantly higher extrahepatic and early recurrence rates after surgical resection. These findings emphasize the importance of assessing the PNI and GNRI for predicting the prognosis of patients with HCC undergoing surgical resection.

The importance of nutritional assessment in patients with HCC has been recognized and various factors have been investigated. We recently reported that malnutrition, as defined by the modified Global Leadership Initiative on Malnutrition (GLIM) criteria, predicted postoperative complications, OS, and RFS in patients with poor liver function [15]. Pinato et al. showed that the PNI was an independent predictor of poor OS in patients with HCC [7]. Chan et al. [14] demonstrated that the PNI was a significant prognostic factor for OS and disease-free survival in patients with very early- or early-stage HCC who underwent curative surgery. Regarding other inflammatory factors, neither the neutrophil-to-lymphocyte ratio nor the platelet-to-lymphocyte ratio showed any prognostic significance in the same cohort. In contrast, nutritional status using the GNRI revealed that a low GNRI in elderly patients was associated with worse postoperative clinical outcomes, such as liver failure, severe complications, and worse OS, but not RFS [11]. More recently, a combination of the PNI and GNRI predicted the risk of OS and RFS in patients with HCC after surgery [13]. Yang et al. showed that patients with high GNRI and PNI scores had the best long-term prognoses. Similarly, in the present study, patients with a low PNI or GNRI had significantly worse OS and RFS than those with a high PNI and GNRI.

In the present study, a low PNI or GNRI could reflect poor OS in various situations. In the analysis of all cases, male sex, an ICG-R15 > 10%, a tumor size > 30 mm, microvascular invasion, and a PNI < 45 or a GNRI < 98 were independent predictive factors for OS. Among patients with liver dysfunction (ICG-R15 > 10%), a low PNI or GNRI was an independent risk factor for OS. Regarding tumor factors, a low PNI or GNRI was an independent risk factor for OS in patients with a large tumor size (>30 mm) and without microvascular invasion. Hepatic functional reserve, tumor size, and vascular invasion are important factors in determining HCC treatment [16]. Our results suggest that the PNI and GNRI may be important predictors independent of these factors.

Recurrence is a major complication in the surgical treatment of patients with HCC. The risk factors for recurrence, the usefulness of adjuvant therapy to prevent recurrence, and the management of recurrence after resection have been investigated previously [17]. However, the prevention and management of recurrence are controversial. According to the literature, men have a higher risk of HCC recurrence than women [18]. Many studies have shown that microvascular invasion is associated with a high incidence of recurrence and poor long-term prognosis [19,20]. Microvascular invasion is a histological feature of HCC related to aggressive biological behavior. However, the diagnosis of microvascular invasion is determined by a histologic examination of the surgical specimens obtained after hepatic resection. Therefore, the influence of the diagnosis on preoperative decision making is limited [21]. Owing to a lack of a specific and practical predictive methods for microvascular invasion, Lei et al. [19] developed a nomogram to predict microvascular invasion presence before liver resection. The preoperative factors associated with microvascular invasion were a large tumor diameter, multiple tumors, an incomplete tumor capsule, higher serum AFP levels, a platelet count less than 100 × 103/µL, a hepatitis B virus DNA load greater than 104 IU/mL, and a typical dynamic pattern of tumors on contrast-enhanced magnetic resonance images. These independently associated risk factors were used to form a microvascular invasion risk estimation nomogram. Regarding preoperative serum inflammatory markers, it has been reported that a higher neutrophil/lymphocyte ratio and a lower PNI are associated with microvascular invasion [22]. In our present study, male sex and microvascular invasion were independent prognostic factors for RFS, but a PNI < 45 or a GNRI < 98 were not independent factors for RFS. This may be because a low PNI or a low GNRI were significantly associated with microvascular invasion.

The prevention and management of recurrence are the most important factors that determine prognosis after liver surgery. The timing and location of recurrence are particularly related to prognosis. Herrero et al. [17] reported that 12 months was the most useful cutoff time period to define early recurrence after surgery from recent larger multicenter studies [23,24]. Early recurrence commonly interacts with tumor-related factors such as AFP, tumor number, tumor size, poor differentiation, and vascular invasion [25,26]. Guo et al. [27] evaluated preoperative factors for predicting early recurrence in patients with HCC who underwent repeat liver resection and showed that an elevated platelet-to-lymphocyte ratio >103.6 and AFP ≥ 200 were independent predictors of early recurrence. In the present study, the early recurrence rate was significantly higher in patients with a low PNI or GNRI. Moreover, both the PNI and GNRI were significantly lower in patients with early recurrence than in those without early recurrence. Patients with a low PNI or GNRI may not live long enough to observe late recurrence, and this may have influenced the results. Assessment of the PNI and GNRI may be an important preoperative tool for predicting early recurrence.

The recurrence pattern is another important factor that affects long-term prognosis and treatment strategy. In cases of extrahepatic recurrence, surgical resection is difficult, and treatment options are limited. Therefore, our results are clinically important because extrahepatic recurrence was significantly higher in patients with a low PNI or GNRI. Recently, Hirokawa et al. [28] showed that the GNRI was a prognostic factor for patients with HCC treated with atezolizumab plus bevacizumab. Thus, low PNI and GRNI values may affect treatment after recurrence and are important prognostic factors.

The present study has several limitations. First, this was a retrospective, single-center study, and larger prospective studies are needed to confirm and update our conclusions. Second, there was a selection bias in the study; hepatectomy and radiofrequency ablation are now equally recommended for up to three HCCs ≤ 3 cm in size in the treatment algorithm. This study did not include patients who underwent non-surgical treatment. Further studies that include patients who underwent both surgical and non-surgical treatment are needed. Third, treatment after recurrence was not analyzed, and further studies are necessary to confirm treatment and survival after recurrence. Despite these limitations, the present study demonstrates the probability that the PNI and GNRI are useful indicators of prognosis in patients with HCC after surgery.

## 5. Conclusions

In conclusion, our study demonstrates that low PNI and GNRI scores are significant predictors of poor prognosis in patients with HCC undergoing surgical resection. It is therefore worth noting that there is a significant association between nutritional index and recurrence pattern.

## Figures and Tables

**Figure 1 nutrients-16-00940-f001:**
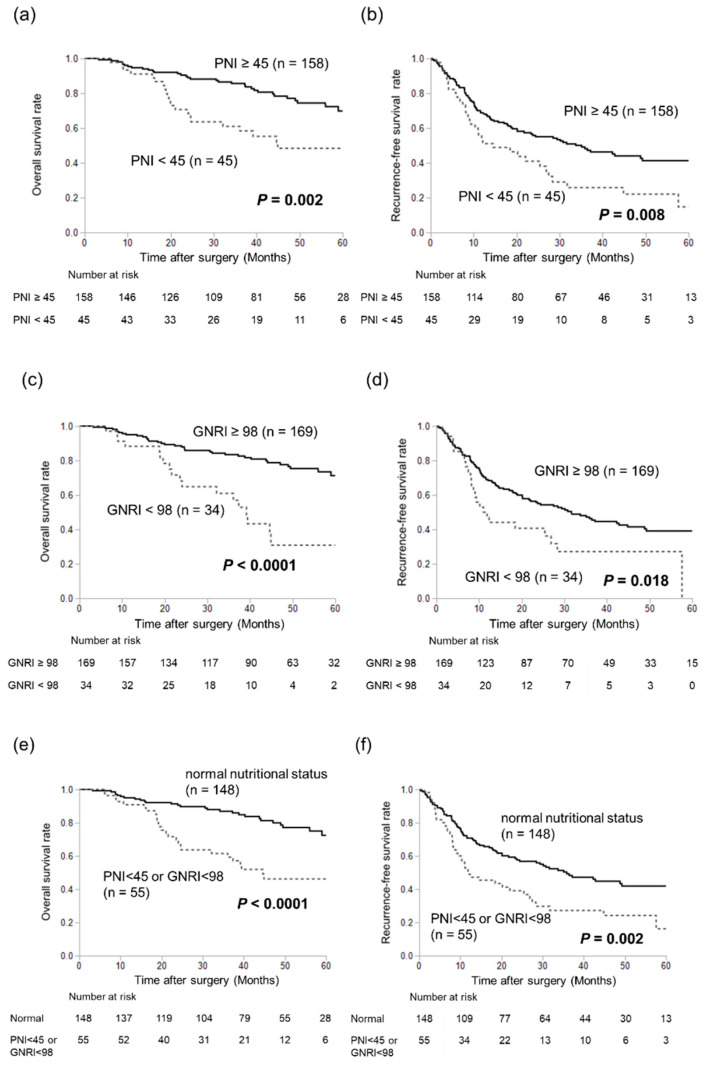
Kaplan–Meier survival plots comparing overall survival and recurrence-free survival for patients stratified as nutritional index. Patients with a low PNI had significantly worse OS (**a**) and RFS (**b**) than those with a high PNI. Patients with a low GNRI had significantly worse OS (**c**) and RFS (**d**) than those with a high GNRI. Patients with a low PNI or GNRI had significantly worse OS (**e**) and RFS (**f**).

**Figure 2 nutrients-16-00940-f002:**
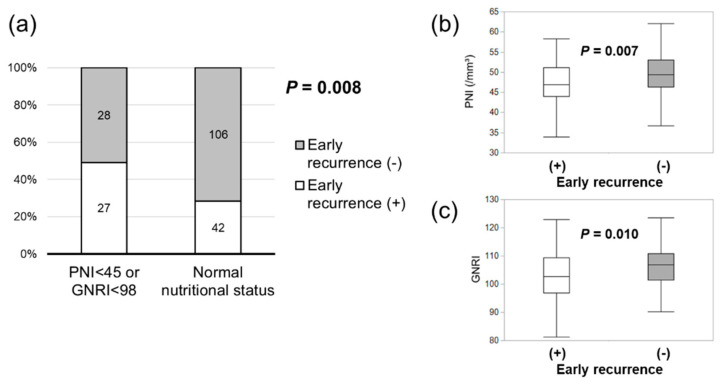
Association between the nutritional index and early recurrence. Early recurrence was significantly higher in patients with a low PNI or GNRI (**a**). Both the PNI (**b**) and GNRI (**c**) were significantly lower in patients with early recurrence.

**Figure 3 nutrients-16-00940-f003:**
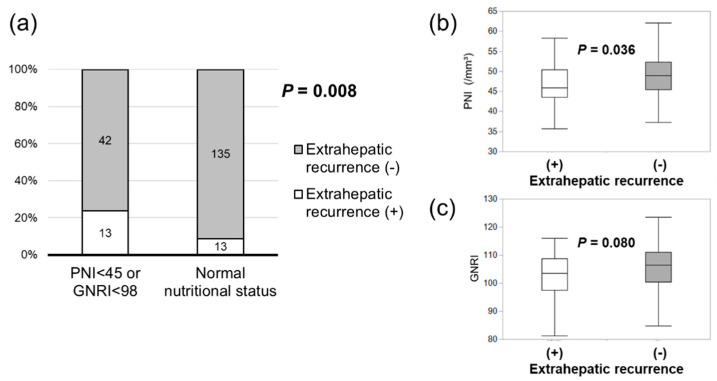
Association between the nutritional index and extrahepatic recurrence. Extrahepatic recurrence was significantly higher in patients with a low PNI or GNRI (**a**). Both the PNI (**b**) and GNRI (**c**) were significantly lower in patients with extrahepatic recurrence.

**Table 1 nutrients-16-00940-t001:** Clinical characteristics of patients in the two groups classified according to the PNI or GNRI.

	PNI			GNRI		
Variables	≥45 (*n* = 158)	<45 (*n* = 45)	*p*-Value	≥98 (*n* = 169)	<98 (*n* = 34)	*p*-Value
Host-related factors						
Age (years)	72 (18–88)	71 (51–86)	0.368	72 (18–88)	71 (51–87)	0.798
Sex: Male	132 (84%)	354 (76%)	0.273	139 (82%)	27 (79%)	0.636
BMI (kg/m^2^)	23.3 (17.3–34.3)	22.1 (17.0–27.3)	0.009 *	23.6 (17.4–34.3)	20.7 (17.0–25.7)	<0.001 *
Skeletal muscle mass index (cm^2^/m^2^)	39.3 (20.7–72.2)	35.5 (22.5–47.7)	0.004 *	39.6 (20.7–72.2)	34.7 (22.5–48.7)	<0.001 *
Hand grip strength (kg)	33.0 (8.6–49.9)	28.0 (14.6–49.9)	0.039 *	32.8 (8.6–49.9)	29.1 (22.0–45.2)	0.229
Etiology HBV/HCV/NBNC	20/56/82	2/22/21	0.114	19/68/82	3/10/21	0.444
Platelet count (/µL)	16.7 (5.7–57.7)	14.5 (5.6–51.7)	0.346	16.2 (5.7–54.3)	17.3 (5.6–57.7)	0.207
Lymphocytes (/µL)	1580 (700–3410)	1030 (340–2340)	<0.001 *	1500 (520–3410)	1380 (340–2590)	0.249
PT (%)	95 (11–121)	89 (65–116)	0.077	95 (11–121)	90 (65–116)	0.142
Total bilirubin (mg/dL)	0.8 (0.3–3.1)	0.8 (0.2–1.6)	0.200	0.8 (0.3–3.1)	0.8 (0.2–2.0)	0.833
Albumin (mg/dL)	4.2 (3.3–5.3)	3.6 (2.8–4.1)	<0.001 *	4.2 (3.3–5.3)	3.5 (2.8–4.1)	<0.001 *
CRP (mg/dL)	0.09 (0.01–8.93)	0.27 (0.01–7.51)	<0.001 *	0.08 (0.01–8.93)	0.33 (0.02–8.37)	<0.001 *
ICG-R15 (%)	15.4 (1.6–91.8)	14.6 (1.5–52.3)	0.557	15.4 (1.6–91.8)	13.5 (1.5–52.3)	0.765
Child-Pugh Score A	156 (99%)	41 (91%)	0.023 *	168 (99%)	29 (85%)	<0.001 *
AFP (ng/mL)	7.6 (1.2–108,317)	41.6 (1.0–275,819)	0.025 *	7.6 (1.0–108,317)	22.8 (1.0–275,819)	0.112
Operative procedures						
Anatomical	88 (56%)	29 (64%)	0.311	93 (55%)	24 (71%)	0.128
Operation time (min)	331 (105–643)	362 (150–682)	0.020 *	331 (105–643)	368 (150–682)	0.059
Blood loss (mL)	114 (0–2050)	312 (8–7219)	<0.001 *	126 (0–2258)	327 (8–7219)	<0.001 *
Postoperative hospitalization (days)	10 (5–88)	13 (8–196)	<0.001 *	11 (5–141)	14 (8–196)	<0.001 *
Complications (Clavien–Dindo grade ≥ 3)	18 (11%)	10 (22%)	0.085	20 (12%)	8 (24%)	0.093
Tumor-related factors						
Tumor size (mm)	3.0 (0.7–22.0)	5.0 (1.1–16.0)	<0.001 *	3.0 (0.7–22.0)	7.0 (1.5–17.0)	<0.001 *
Multiple tumors	29 (18%)	9 (20%)	0.830	34 (20%)	4 (12%)	0.338
Poor differentiation	31 (20%)	5 (11%)	0.268	29 (17%)	7 (21%)	0.620
Microvascular invasion (+)	57 (36%)	25 (56%)	0.025 *	64 (38%)	18 (53%)	0.126

Data are expressed as the median (interquartile range) or number of patients (%). * *p* value < 0.05. Abbreviations: PNI, prognostic nutritional index; GNRI, geriatric nutritional risk index; BMI, body mass index; HBV, hepatitis B virus; HCV, hepatitis C virus; NBNC, non-B non-C; PT, prothrombin time; CRP, C-reactive protein; ICGR-15, indocyanine green retention rate at 15 min; AFP, alpha-fetoprotein.

**Table 2 nutrients-16-00940-t002:** Univariate/multivariate analysis for overall survival.

Variables	Univariate Analysis			Multivariate Analysis		
	HR	95%CI	*p*-Value	HR	95%CI	*p*-Value
Age > 80 (years)	1.93	0.99–3.77	0.053			
Sex: male	7.10	1.73–29.16	0.007 *	9.85	2.37–41.00	0.002 *
HCV	1.04	0.60–1.79	0.883			
Skeletal muscle loss	1.21	0.68–2.14	0.524			
Child-Pugh Score B or C	1.66	0.40–6.83	0.485			
AFP > 40 (ng/mL)	1.56	0.92–2.67	0.101			
ICG-R15 > 10 (%)	2.51	1.07–5.86	0.034 *	3.48	1.42–8.50	0.006 *
PNI < 45	2.35	1.36–4.05	0.002 *			
GNRI < 98	3.26	1.84–5.77	<0.001 *			
PNI < 45 or GNRI < 98	2.78	1.63–4.72	<0.001 *	2.21	1.21–4.05	0.010 *
Operation time > 300 (min)	2.45	1.23–4.86	0.010 *	1.05	0.48–2.27	0.903
Blood loss > 500 (mL)	2.19	1.17–4.10	0.014 *	1.73	0.87–3.42	0.116
Complications (Clavien–Dindo grade ≥ 3)	1.88	1.01–3.50	0.048 *	1.21	0.63–2.35	0.562
Tumor size > 30 (mm)	1.96	1.12–3.41	0.018 *	1.95	1.06–3.61	0.033 *
Multiple tumors	1.64	0.88–3.06	0.122			
Poor differentiation	1.43	0.74–2.48	0.290			
Microvascular invasion (+)	3.23	1.86–5.62	<0.001 *	2.87	1.58–5.24	<0.001 *

Abbreviations: HR, hazard ratio; CI, confidence interval; AFP, alpha-fetoprotein; ICGR-15, indocyanine green retention rate at 15 min; PNI, prognostic nutritional index; GNRI, geriatric nutritional risk index. * *p* value < 0.05.

**Table 3 nutrients-16-00940-t003:** Subgroup analysis for overall survival.

Variables	Multivariate Analysis		
	HR	95%CI	*p*-Value
(a) ICGR15 > 10 (%)			
Sex: male	8.64	2.06–36.25	0.003 *
PNI < 45 or GNRI < 98	2.56	1.33–4.95	0.005 *
Operation time > 300 (min)	0.83	0.36–1.91	0.663
Blood loss > 500 (mL)	2.28	1.12–4.63	0.023 *
Complications (Clavien–Dindo grade ≥ 3)	1.05	0.53–2.10	0.888
Tumor size > 30 (mm)	2.47	1.30–4.69	0.006 *
Microvascular invasion (+)	3.38	1.76–6.49	<0.001 *
(b) Tumor size > 30 (mm)			
Sex: male	6.15	1.46–25.98	0.014 *
ICG-R15 > 10 (%)	4.61	1.61–13.24	0.005 *
PNI < 45 or GNRI < 98	2.53	1.25–5.10	0.010 *
Operation time > 300 (min)	2.30	0.53–10.01	0.267
Microvascular invasion (+)	3.07	1.40–6.71	0.005 *
(c) Microvascular invasion (+)			
Sex: male	12.23	1.67–89.80	0.014 *
ICG-R15 > 10 (%)	3.96	1.20–13.02	0.024 *
(d) Microvascular invasion (−)			
Sex: male	4.24	0.56–32.11	0.162
ICG-R15 > 10 (%)	1.93	0.52–7.13	0.323
PNI < 45 or GNRI < 98	4.10	1.63–10.30	0.003 *
Tumor size > 30 (mm)	1.38	0.54–3.56	0.499

Abbreviations: HR, hazard ratio; CI, confidence interval; ICGR-15, indocyanine green retention rate at 15 min; PNI, prognostic nutritional index; GNRI, geriatric nutritional risk index. * *p* value < 0.05.

**Table 4 nutrients-16-00940-t004:** Univariate/multivariate analysis for recurrence-free survival.

Variables	Univariate Analysis			Multivariate Analysis		
	HR	95%CI	*p*-Value	HR	95%CI	*p*-Value
Age > 80 (years)	0.88	0.51–1.55	0.668			
Sex: male	2.57	1.41–4.69	0.002 *	3.00	1.64–5.50	<0.001 *
HCV	1.10	0.75–1.60	0.619			
Skeletal muscle loss	0.79	0.54–1.15	0.221			
Child-Pugh Score B or C	0.95	0.30–3.01	0.936			
AFP > 40 (ng/mL)	1.19	0.81–1.75	0.363			
ICG-R15 > 10 (%)	1.45	0.90–2.33	0.128			
PNI < 45	1.73	1.15–2.59	0.008 *			
GNRI < 98	1.71	1.09–2.69	0.020 *			
PNI < 45 or GNRI < 98	1.81	1.23–2.66	0.003 *	1.40	0.92–2.13	0.115
Operation time > 300 (min)	1.39	0.94–2.07	0.101			
Blood loss > 500 (mL)	2.10	1.29–3.40	0.003 *	1.66	0.99–2.76	0.054
Complications (Clavien–Dindo grade ≥ 3)	1.24	0.75–2.05	0.408			
Tumor size > 30 (mm)	1.50	1.03–2.17	0.034 *	1.29	0.88–1.90	0.197
Multiple tumors	1.38	0.87–2.17	0.172			
Poor differentiation	1.26	0.79–2.00	0.334			
Microvascular invasion (+)	2.24	1.55–3.24	<0.001 *	2.05	1.40–3.01	<0.001 *

Abbreviations: HR, hazard ratio; CI, confidence interval; AFP, alpha-fetoprotein; ICGR-15, indocyanine green retention rate at 15 min; PNI, prognostic nutritional index; GNRI, geriatric nutritional risk index. * *p* value < 0.05.

## Data Availability

The original contributions presented in the study are included in the article, further inquiries can be directed to the corresponding author.

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
