# Peer review of "Lower Geriatric Nutritional Risk Index and Prognostic Nutritional Index Predict Postoperative Prognosis in Patients with Hepatocellular Carcinoma"

_nutrients, 2024, doi:10.3390/nu16070940_

Round 1

Reviewer 1 Report

Comments and Suggestions for Authors

The authors conducted an observational study to examine the associations of preoperative geriatric nutritional risk index and prognostic nutritional index with postoperative survival and recurrence in patients with hepatocellular carcinoma. By recruiting a total of 203 hepatocellular carcinoma patients admitted to a hospital in Japan, the authors showed that having a low preoperative prognostic nutritional index or geriatric nutritional risk index was associated with lower overall survival, particularly among those with liver dysfunction or larger tumors. The organization of the manuscript is clear. There are some comments.

1.     Materials and Methods: Please provide more details on the study sample recruitment method, including eligibility criteria and participant selection methods (for instance, consecutive patients admitted to the department). If the information is available, I also recommended that the authors present the number of potentially eligible participants, the number of potentially eligible participants assessed for eligibility, the number of confirmed eligible, and the number of confirmed eligible and agreed to participate. I recommend appropriate citations here if methodological details have been reported in previous literature.

2.     Materials and Methods: The authors applied the Cox proportional hazards models. Please describe whether the proportional hazards assumption was tested and whether the assumption was met.

3.     Results (Figure 1): Kaplan–Meier curves were shown. Please also present the number at risk for each group at the bottom of the figure along the x-axis.

4.     Results (Table 3): Results of subgroup analysis for overall survival were presented. Please also show the results for those with ICGR15<=10% and those with tumor size <=30mm.

5.     Discussion: The authors also observed that patients with early recurrence had a lower preoperative prognostic nutritional index or geriatric nutritional risk index than those with late recurrence. This difference may arise just because patients with a low preoperative prognostic nutritional index or geriatric nutritional risk index could not live long enough to allow the late recurrence to be observed. A discussion of this issue is recommended.

6.     Discussion: After adjusting for other factors, the authors failed to observe a statistically significant association between the preoperative geriatric nutritional risk index and the prognostic nutritional index and recurrence-free survival (Table 4). I recommend a discussion of this finding. For instance, could it be possible that some of the other factors that were also included in the multivariate model actually mediated the association of low preoperative prognostic nutritional index or geriatric nutritional risk index with recurrence-free survival?

7.     Abstract: “Low PNI or GNRI scores were significantly associated with early and extrahepatic recurrences after surgical resection.” However, the association was not adjusted for other factors (Figures 2 and 3). I recommend re-writing the sentence. A more appropriate statement could be, for instance, “Patients with early recurrence had lower PNI or GNRI scores than those with late recurrence.”

8.     Abstract: “The PNI and GNRI are effective tools for predicting the prognosis and recurrence patterns of patients with HCC after hepatic resection.” A more appropriate conclusion could be “The PNI and GNRI might be useful in predicting the prognosis and recurrence patterns of patients with HCC after hepatic resection.”

Author Response

Response to Reviewer 1 Comments

1. Summary

Thank you very much for taking the time to review this manuscript. Please find the detailed responses below and the corresponding revisions highlighted in the re-submitted files.

2. Questions for General Evaluation

Reviewer’s Evaluation

Response and Revisions

Does the introduction provide sufficient background and include all relevant references?

Yes

Are all the cited references relevant to the research?

Yes

Is the research design appropriate?

Yes

Are the methods adequately described?

Can be improved

We revised the manuscript.

Are the results clearly presented?

Can be improved

We revised the manuscript.

Are the conclusions supported by the results?

Yes

3. Point-by-point response to Comments and Suggestions for Authors

Comments 1: Materials and Methods: Please provide more details on the study sample recruitment method, including eligibility criteria and participant selection methods (for instance, consecutive patients admitted to the department). If the information is available, I also recommended that the authors present the number of potentially eligible participants, the number of potentially eligible participants assessed for eligibility, the number of confirmed eligible, and the number of confirmed eligible and agreed to participate. I recommend appropriate citations here if methodological details have been reported in previous literature.

Response 1: We appreciate your bringing this important point to our attention. This study included consecutive 203 patients with HCC who underwent initial hepatic resection between January 2016 and September 2022 at our department. We have now added this information to the Materials and Methods section.

Materials and Methods:

We retrospectively analyzed 203 consecutive patients with HCC who underwent initial hepatic resection between January 2016 and September 2022 at the Department of Hepatobiliary and Pancreatic Surgery, Gunma University Hospital.

(Page 3, lines 59–61)

Comments 2: Materials and Methods: The authors applied the Cox proportional hazards models. Please describe whether the proportional hazards assumption was tested and whether the assumption was met.

Response 2: We appreciate that you were able to point out this important fact. We graphically evaluated the proportional hazards assumption with Kaplan-Meier curves and log-log plots, and the assumption was met.

Comments 3: Results (Figure 1): Kaplan–Meier curves were shown. Please also present the number at risk for each group at the bottom of the figure along the x-axis.

Response 3: We appreciate this important comment. As you pointed out, we added the number at risk for each group at the bottom of the figure 1.

Comments 4: Results (Table 3): Results of subgroup analysis for overall survival were presented. Please also show the results for those with ICGR15<=10% and those with tumor size <=30mm.

Response 4: We appreciate this important comment. We also did subgroup analysis for overall survival (OS) with ICGR15 ≤10% and tumor size ≤30mm. In subgroup analysis of the patients with ICG R15 ≤10%, there was no significant factor for OS in univariate analysis. Furthermore, in subgroup analysis of the patients with tumor size ≤30mm, microvascular invasion was the only independent prognostic factor in univariate analysis for OS. We added these results to the Results section.

Results:

Low PNI or low GNRI was an independent risk factor for OS in subgroup analysis of the patients with ICG R15>10%, tumor size>30 mm, and without microvascular invasion (Table 3). In subgroup analysis of the patients with ICG R15 ≤10%, there was no significant factor for OS in univariate analysis. Furthermore, in subgroup analysis of the patients with tumor size ≤30mm, microvascular invasion was the only independent prognostic factor in univariate analysis for OS.

(Page 6, lines 160–163)

Comments 5: Discussion: The authors also observed that patients with early recurrence had a lower preoperative prognostic nutritional index or geriatric nutritional risk index than those with late recurrence. This difference may arise just because patients with a low preoperative prognostic nutritional index or geriatric nutritional risk index could not live long enough to allow the late recurrence to be observed. A discussion of this issue is recommended.

Response 5: Thank you for providing these insightful comments. In the study shown in Figure 2, all cases were analyzed, including not only patients with recurrence but also patients without recurrence. As you pointed out, patients with a low PNI or GNRI may not live long enough to observe late recurrence. We mentioned this additionally.

Discussion:

In the present study, the early recurrence rate was significantly higher in patients with a low PNI or GNRI. Moreover, both the PNI and GNRI were significantly lower in patients with early recurrence than in those without early recurrence. Patients with a low PNI or GNRI may not live long enough to observe late recurrence, and this may have influenced the results.

(Page 10, lines 272–273)

Comments 6: Discussion: After adjusting for other factors, the authors failed to observe a statistically significant association between the preoperative geriatric nutritional risk index and the prognostic nutritional index and recurrence-free survival (Table 4). I recommend a discussion of this finding. For instance, could it be possible that some of the other factors that were also included in the multivariate model actually mediated the association of low preoperative prognostic nutritional index or geriatric nutritional risk index with recurrence-free survival?

Response 6: We appreciate this important comment. As you pointed out, the low PNI or low GNRI group was significantly correlated with microvascular invasion. It is possible that this factor influenced the results. Microvascular invasion is an important prognostic factor and we added consideration to the Discussion section.

Discussion:

Many studies have shown that microvascular invasion is associated with a high incidence of recurrence and poor long-term prognosis [19,20]. Microvascular invasion is a histological feature of HCC related to aggressive biological behavior. However, the diagnosis of microvascular invasion is determined on histologic examination of the surgical specimens obtained after hepatic resection. Therefore, the influence of the diagnosis on preoperative decision making is limited [21]. Owing to lack of a specific and practical predictive method of microvascular invasion, Lei et al. [19] developed a nomogram to predict microvascular invasion presence before liver resection. The preoperative factors associated with microvascular invasion were large tumor diameter, multiple tumors, incomplete tumor capsule, higher serum AFP level, platelet count less than 100 × 103/µL, hepatitis B virus DNA load greater than 104 IU/mL, and a typical dynamic pattern of tumors on contrast-enhanced magnetic resonance imaging. These independently associated risk factors were used to form a microvascular invasion risk estimation nomogram. Regarding preoperative serum inflammatory markers, it has been reported that higher neutrophil lymphocyte ratio and lower PNI were associated with microvascular invasion [22]. In our present study, male sex and microvascular invasion were independent prognostic factors for RFS, but PNI<45 or GNRI<98 was not an independent factor for RFS. This may be because low PNI or low GNRI was significantly associated with microvascular invasion.

(Page 9, lines 244–260)

References

21.         Rodriguez-Peralvarez, M.; Luong, T. V.; Andreana, L.; Meyer, T.; Dhillon, A. P.; Burroughs, A. K. A systematic review of microvascular invasion in hepatocellular carcinoma: diagnostic and prognostic variability. Ann Surg Oncol 2013, 20, 325–339. doi: 10.1245/s10434-012-2513-1.

22.         Xu, X; Sun, S.; Liu, Q.; Liu, X.; Wu, F.; Shen, C. Preoperative application of systemic inflammatory biomarkers combined with MR imaging features in predicting microvascular invasion of hepatocellular carcinoma. Abdom Radiol 2022, 47, 1806–1816.

Comments 7: Abstract: “Low PNI or GNRI scores were significantly associated with early and extrahepatic recurrences after surgical resection.” However, the association was not adjusted for other factors (Figures 2 and 3). I recommend re-writing the sentence. A more appropriate statement could be, for instance, “Patients with early recurrence had lower PNI or GNRI scores than those with late recurrence.”

Response 7: Thank you for this important suggestion. We have revised the text in the Abstract section accordingly.

Abstract:

Patients with early recurrence had lower PNI and GNRI scores than those without early recurrence. Patients with extrahepatic recurrence had lower PNI and GNRI scores than those without extrahepatic recurrence.

(Page 1, lines 21–23)

Comments 8: Abstract: “The PNI and GNRI are effective tools for predicting the prognosis and recurrence patterns of patients with HCC after hepatic resection.” A more appropriate conclusion could be “The PNI and GNRI might be useful in predicting the prognosis and recurrence patterns of patients with HCC after hepatic resection.”

Response 8: Thank you for this important suggestion. We have revised the text in the Abstract section accordingly.

Abstract:

Low PNI or GNRI scores were significantly associated with early and extrahepatic recurrences after surgical resection. The PNI and GNRI might be useful in predicting the prognosis and recurrence patterns of patients with HCC after hepatic resection.

(Page 1, lines 23–24)

4. Response to Comments on the Quality of English Language

Point 1: English language fine. No issues detected

Response 1: Thank you very much.

5. Additional clarifications

None.

Reviewer 2 Report

Comments and Suggestions for Authors

The authors present a retrospective study regarding the influence of PNI and GNRI on disease recurrence in patients with HCC undergoing surgery.  

For the introduction, I would detail more about HCC.

I would add some information regarding the possible viral B and C markers and the association between them, PNI, GNRI, and recurrence. And also if patients undergo antiviral treatment.

Author Response

Response to Reviewer 2 Comments

1. Summary

Thank you very much for taking the time to review this manuscript. Please find the detailed responses below and the corresponding revisions highlighted in the re-submitted files.

2. Questions for General Evaluation

Reviewer’s Evaluation

Response and Revisions

Does the introduction provide sufficient background and include all relevant references?

Can be improved

We revised the manuscript.

Are all the cited references relevant to the research?

Can be improved

Is the research design appropriate?

Can be improved

Are the methods adequately described?

Can be improved

We revised the manuscript.

Are the results clearly presented?

Can be improved

We revised the manuscript.

Are the conclusions supported by the results?

Yes

3. Point-by-point response to Comments and Suggestions for Authors

Comments 1: For the introduction, I would detail more about HCC.

I would add some information regarding the possible viral B and C markers and the association between them, PNI, GNRI, and recurrence. And also if patients undergo antiviral treatment.

Response 1: We appreciate your bringing this important point to our attention. Viral hepatitis was not significantly associated with PNI and GNRI (Table 1). Univariate and multivariate analyses of OS and RFS showed that HCV positivity was not an independent prognostic factor in this study (Table 2, 3). We have now added some information to the Introduction, Materials and Methods, and Results section.

Introduction:

Hepatic resection is one of the mainstay curative treatments for HCC. Recent advances in perioperative management and strict surgical criteria have enabled safe hepatic resection for HCC. However, patients with HCC show high recurrence rates even after curative surgical resection, and many cases develop an unresectable status [2].

(Page 1, lines 29–31)

Materials and Methods:

The baseline clinical and demographic characteristics and treatment-related details of all patients were collected from their medical records. Positive anti–HCV findings were considered to show that HCC was caused by hepatitis C virus (HCV), whereas HCC due to hepatitis B virus (HBV) was determined when the HBV surface antigen was positive.

(Page 2, lines 67–69)

Results:

3.1. Clinical characteristics of patients in the two groups classified according to PNI and GNRI

Based on this definition, 45 (22%) patients were allotted to the low PNI and 34 (17%) patients were allotted to the low GNRI groups. Basal liver disease was due to HCV in 78 (38.4), HBV in 22 (10.8%), and others in 103 (50.7%) patients. Of the 78 patients with HCV, 54 (69.2%) patients had obtained a sustained virological response by antiviral treatment (interferon or direct acting antiviral treatment) before initial hepatic resection.

(Page 3, lines 111–114)

4. Response to Comments on the Quality of English Language

Point 1: I am not qualified to assess the quality of English in this paper.

5. Additional clarifications

None.
